# Molecular Epidemiology of Group A *Streptococcus* Infections in The Gambia

**DOI:** 10.3390/vaccines9020124

**Published:** 2021-02-04

**Authors:** Sona Jabang, Annette Erhart, Saffiatou Darboe, Aru-Kumba Baldeh, Valerie Delforge, Gabriella Watson, Ebenezer Foster-Nyarko, Rasheed Salaudeen, Bolarinde Lawal, Grant Mackenzie, Anne Botteaux, Martin Antonio, Pierre R. Smeesters

**Affiliations:** 1Medical Research Council Unit The Gambia at the London School of Hygiene and Tropical Medicine, Banjul 273, The Gambia; sjabang@mrc.gm (S.J.); sdarboe@mrc.gm (S.D.); arkbaldeh@mrc.gm (A.-K.B.); Gabriella.watson@uhs.nhs.uk (G.W.); Ebenezer.Foster-Nyarko@quadram.ac.uk (E.F.-N.); rsalaudeen@mrc.gm (R.S.); blawal@mrc.gm (B.L.); gmackenzie@mrc.gm (G.M.); mantonio@mrc.gm (M.A.); 2Laboratoire de Génétique et Physiologie Bactérienne, IBMM, Université Libre de Bruxelles, 12 Rue des Professeurs Jenner et Brachet, 6041 Gosselies, Belgium; valerie.delforge@ulb.ac.be (V.D.); anne.botteaux@ulb.be (A.B.); pierre.smeesters@huderf.be (P.R.S.); 3Southampton University Hospital, Southampton SO16 6YD, UK; 4Department of Pediatrics, Hôpital Universitaire des Enfants Reine Fabiola, Université Libre de Bruxelles, 1050 Brussels, Belgium

**Keywords:** group A *streptococcus*, *emm*-type, *emm*-cluster, vaccine, The Gambia

## Abstract

Molecular epidemiological data on Group A *Streptococcus* (GAS) infection in Africa is scarce. We characterized the *emm-*types and *emm*-clusters of 433 stored clinical GAS isolates from The Gambia collected between 2004 and 2018. To reduce the potential for strain mistyping, we used a newly published primer for *emm*-typing. There was considerable strain diversity, highlighting the need for vaccine development offering broad strain protection.

## 1. Introduction

Group A *Streptococcus* (GAS) causes a significant morbidity and mortality burden globally due to a variety of clinical manifestations and subsequent immunologically mediated complications, including acute rheumatic fever (ARF) and rheumatic heart disease (RHD) [1]. The highest burden of disease is found in low- and middle-income countries (LMIC) although there may be considerable geographic variability [1]. Even though there has long been a need for a GAS vaccine, recent advances are favoring its development. In 2018, the World Health Assembly supported GAS vaccine development through a renewed action to control both ARF and RHD [2]. A global atlas of vaccine candidate antigens, originating from a genetically diverse worldwide study of 2083 GAS genomes, was also recently published [3]. Finally, a human challenge model of GAS acute throat infection has been established and may enhance vaccine development [4]. The surface M protein is a major virulence determinant, and its N-terminal amino-acid residue consists of a highly variable amino-acid sequence that results in significant antigenic diversity (*emm*-types). This was the basis for a recombinant hybrid vaccine containing M protein epitopes from multiple GAS serotypes (i.e., the current 30-valent vaccine). The latter, together with another vaccine targeting the conserved J8 region of the M protein, has so far reached phase 1 of clinical trials [2]. The 30-valent vaccine covers the most frequent serotypes circulating in high-income countries (HIC), but concerns have been raised about its coverage in LMIC settings where the diversity of GAS *emm*-types, and therefore M serotypes, is greater [5,6]. However, the pre-clinical development of the 30-valent vaccine has demonstrated in vitro cross-opsonization against isolates expressing M proteins that are not included in the vaccine, suggesting that its coverage could be higher than expected [7]. The J8 vaccine candidate was designed to offer protection against most GAS isolates by using a relatively conserved vaccine antigen [2].

Importantly, only limited epidemiological data are currently available from many regions, making vaccine coverage estimates imprecise. In Africa, GAS molecular data (*emm*-sequence types) are reported from only five countries (Tunisia, Mali, Ethiopia, Kenya, and South Africa), as recently reviewed [8]. The aim of the current study was to characterize the molecular epidemiology of GAS infections in The Gambia, West Africa, and to assess the theoretical coverage of the 30-valent vaccine candidate.

## 2. Material and Methods

The clinical microbiology laboratory database at the Medical Research Council The Gambia (MRCG) at the London School of Hygiene and Tropical Medicine was interrogated to identify all clinical GAS isolates recorded between December 2004 and June 2018. Invasive GAS isolates identified within the Pneumococcal Surveillance Project (PSP) (https://www.mrc.gm/pneumococcal-surveillance-project-psp-press-briefing/) between 2008 and 2017 were also included. An invasive GAS disease was defined as the isolation of GAS from a normally sterile body site including bacteremia/septicaemia, meningitis, pneumonia, septic arthritis, osteomyelitis, cellulitis with septicaemia, necrotizing fasciitis, and streptococcal toxic shock syndrome. A non-invasive GAS disease included throat and skin infections without septicaemia.

Stored GAS isolates were retrieved from the MRCG Biobank, sub-cultured and confirmed using a rapid latex streptococcal grouping kit, Streptex (Remel) at the MRCG Clinical Microbiology Laboratory, then sent to the Molecular Bacteriology Laboratory, Brussels (MBLB), Belgium, for genotyping. The MRCG Clinical Microbiology Laboratory is accredited to ISO 15189:2012.

At the MBLB, all isolates were reconfirmed as GAS by colony morphology, beta-hemolysis on 5% sheep blood agar, negative catalase reaction, and detection of Lancefield group A antigen by latex agglutination (Pastorex^TM^ Strep, Biorad, Belgium). These GAS isolates were typed using the recently published updated *emm*-typing protocol [9]. The use of this new PCR-based typing protocol improves the specificity of the *emm*-typing PCR reaction using a primer called CDC3. This method therefore reduces the amplification of multiple bands by PCR, avoiding the misclassification of strains into types based on non-*emm* gene sequences [9]. In addition, we used an *emm*-cluster typing system that classifies the numerous GAS *emm*-types into 48 discrete *emm*-clusters containing closely related M proteins that share binding and structural properties [10]. The 30-valent vaccine coverage was estimated using the latest cross-opsonization data [7,10]. The GAS strain diversity was assessed by Simpson’s reciprocal index. The Gambia Government/MRCG Joint Ethics Committee gave ethical approval for the conduct of the study (SCC 1567-L2018.41).

## 3. Results

Four hundred and thirty-three GAS isolates were identified from 431 patients, of whom 398 presented to the MRCG outpatient department in Fajara (coastal area) between 10 December 2004 and 30 June 2018, and 33 were from the PSP study. The latter were all from children less than five years of age (median 19 months (IQR: 15–32)) with a sex ratio M/F of 1:4, while the former included children and adults aged from <1 month to 77 years (median 13 years (IQR: 2–28)) with a sex ratio of 1:1. Notably, age and sex information were only available for 152 (37.7%) and 188 (46.6%), respectively, of the 398 MRCG Fajara patients. All GAS isolates from the PSP study were from patients with bacteremia, 12 of whom also had pneumonia and three had meningitis. Among the 336 MRCG Fajara patients with available clinical data, the majority (230 = 68.5%) had skin infections (mostly pyoderma), another 20.2% (*n* = 69) had ear-nose-throat (ENT) infections including 39 external otitis and 22 pharyngitis, and 7.3% (*n* = 25) had bacteremia.

Eighty *emm*-types were identified among the 433 GAS isolates, encompassing 22 *emm*-clusters. Most *emm*-types (76.3%; *n* = 61) belonged to 5 *emm*-clusters, and of those remaining, 13 *emm*-types represented 13 single-type clusters. Twenty-two GAS isolates were considered non-typeable (Table 1). No novel *emm*-types were identified, but eight new *emm*-sequences were submitted to the American Center for Disease Control (CDC) and were identified as new subtypes (Table 1). Those new sequences have been included in the worldwide database (https://www.cdc.gov/streplab/groupa-strep/index.html). Simpson’s reciprocal index on *emm-*types was 41.6 (95% confidence interval (CI): 36.7–48.1), indicating considerable diversity.

Excluding the non-typeable isolates, the five most frequent *emm*-clusters were, in decreasing order, E6 (22.3%, 95% CI [17.9; 27.4]), followed by E3 (19.0%, 95%CI [14.9; 23.8]), then E4 (16.5%, 95%CI [12.7; 21.0]), and E2 and D4 (each at 7.9%, 95%CI [5.3; 11.1]), encompassing 66% of The Gambia isolates (Table 1). There was evidence for a different *emm*-cluster distribution among non-invasive and invasive infections. Indeed, among invasive infections, the most predominant cluster was E3 (24.1%), followed by E6 (17.2%), then M95 (12.1%), E4 (8.9%), and E2 and M55 (each 6.9%) (exact test for difference between invasive infections (INV) and non invasive infections (NINV), *p*-value = 0.03). Based on published vaccine ELISA data [7,10], 150 isolates (34.6%) were *emm*-types included by the 30-valent vaccine, and an additional 141 isolates (32.6%) might be covered by the vaccine because of in vitro cross-opsonization. Therefore, the potential coverage could be expected to be 67.2% (95% CI 62.8–71.6%) but may be higher, as 29.6% of the isolates belonged to 37 *emm*-types which have not yet been examined for cross-opsonization.

## 4. Discussion

These data are the first to be published on Group A *Streptococcus* molecular epidemiology in The Gambia. We observed a high level of diversity of GAS strains associated with invasive disease, skin, and throat infections. Although eight new subtypes were identified by this study, no new *emm*-types were discovered, suggesting that the CDC reference laboratory may have good coverage in terms of *emm*-type diversity. The *emm*-type in Simpson’s reciprocal index was 41.6 indicating considerable diversity, a result similar to that seen in other African studies [8]. The predominant *emm*-clusters identified in our study were similar to those reported in the recently published systematic review that included five African countries [8] but were different from high income countries such as the United States, where E4, AC3, and AC4 were the most predominant clusters [11]. Furthermore, the different *emm*-cluster distributions found among non-invasive infections, with a predominance of E6, and invasive infections, with a predominance of E3 followed by E6, were also reported in other African studies [8]. Additionally, two single-type clusters, M55 and M95, that were specifically found in Mali [8] were also relatively common in our study (respectively, *n* = 10 and 20 isolates), suggesting that similarity in GAS circulation may exist. Non-typable isolates are likely to be related to limitations in the *emm*-typing methods [9] or, rather exceptionally, be associated with *emm*-negative strains [3].

Over two third of our isolates (including non-typeables (NTs)) were potentially covered by the current 30-valent vaccine because of potential cross-opsonization, and this figure could be higher if the 37 uninvestigated *emm*-types show cross-opsonization. This highlights the importance and urgent need to further investigate the potential for cross-opsonization for the pending 37 *emm*-types to better estimate the coverage of the 30-valent vaccine both in invasive and non-invasive isolates in The Gambia. A review of African studies showed a potential coverage of 80% of isolates after including those potentially covered by the vaccine as a result of cross-opsonization; however, the coverage would have been only 56% if the *emm-*types covered by the vaccine were taken into account [8]. On the other hand, cross-opsonization in vitro still needs to be assessed in human studies, hence these results must be interpreted cautiously. Vaccine antigen development should definitively aim for the broadest strain coverage possible.

The main limitation of this study was its retrospective design, which was prone to data incompleteness, lack of accuracy, as well as methodological bias. Indeed, our data suffered from a substantial amount of missing and incomplete information on socio-demographic and clinical presentations, which prevented any meaningful analysis of the *emm*-cluster distribution by age, sex, geographical location, or clinical presentation. Going forward, the MRC Clinical Services has now instituted an electronic data capture system to avoid such problems in the future. In addition, compared with the vast majority of skin samples, acute throat infections were likely largely under-represented in our study since they are rarely considered worthy of seeking medical treatment in The Gambia [12]. This could have negatively affected the frequency of some *emm*-types specific to this presentation, if any. Further community-based studies on the molecular epidemiology of Strep A pharyngitis in The Gambia are needed to provide more insights on this aspect. Nevertheless, the fact that our molecular typing results are very similar to those published by a systematic review of eight prospective studies across Africa, including different age groups and presentations, supports their wider validity.

Overall, these data indicate a high diversity of circulating GAS strains in The Gambia, and the urgent need for complementary studies to assess the potential coverage of the 30-valent vaccine candidate. These results highlight the need for robust regional and country-level data to inform future vaccine design. Moving forward, a prospective and comprehensive GAS infection surveillance in The Gambia and in Africa would be highly desirable.

## Figures and Tables

**Table 1 vaccines-09-00124-t001:** *emm*-clusters, *emm-*subtypes characterized in this study (*n* = 433 Group A *Streptococcus* (GAS) isolates), and the number of isolates by non-invasive and invasive infections (*n* = 371).

*emm-*Cluster	GAS Isolates (*n*)	*emm*-Types (*n*)	NINV *n* (%) *n* = 313	INV *n* (%) *n* = 58
E6	88	11.0 # (13), 42.0& (1), 42.3 & (4), 63.0 & (3), 65.0 & (1), 65.4 & (5), 65.7 & (22), 75.1 # (7), 75.3 # (8), 81.2 # (10), 85.1 & (7), 158.0 & (2), 182.1 (5).	67 (21.2)	10 (17.2)
E3	75	25.1& (11), 44.0 # (6), 49.0 # (8), 49.1 # (2), 49.10 # (2), 49.3 # (4), 58.0 # (4), 58.1 # (1), 58.17 # (1), 82.6 # (4), 103.0 (4), 118.0 # (12), 180.0 & (1), 183.2 & (8), 209.3 (7)	50 (15.8)	14 (24.1)
E4	64	8.3 & (2), 28.5 # (17), 73.0 # (2), 77.0 # (15), 87.9 # (3), 88.8 * (1), 88.9 *(4), 89.14 # (4), 89.8 # (2), 109.1 & (9), 169.1 & (4), 232.1 (1)	49 (15.7)	5 (8.6)
E2	31	50.0 (4), 66.1 & (2), 68.0 & (1), 90.1 (3), 92.0 # (3), 104.0 (6), 106.0 (1), 110.10 (1), 166.1 (1), 166.4 (5) 168.1 & (4)	23 (7.3)	4 (6.9)
D4	31	53.1 & (3), 56.3 * (1), 80.0 (2), 86.2 (3), 93.0 (7), 119.2 (2), 192.0 (1), 223.0 (2), 225.0 (3), 230.1 (7)	24 (7.6)	3 (5.2)
M95	20	95.0 & (20)	10 (3.2)	7 (12.1)
E1	13	4.0 # (1), 4.21 # (3), 4.5 # (4), 60.7 & (4), 78.6 *#(1)	10 (3.2)	0
M55	10	55.0 (10)	4 (1.3)	4 (6.9)
D2	9	71.0 & (2), 71.1 & (3), 100.3 & (4)	8 (2.5)	1 (1.7)
M19	8	19.9 # (6), 19.14 *# (1), 19.15 *# (1)	7 (2.2)	1 (1.7)
M74	9	74.0 & (9)	9 (2.8)	0
M57	6	57.0 (6)	3 (1.0)	0
M18	5	18.12 # (1), 18.21 # (4)	2 (0.6)	0
M122	4	122.0 & (4)	4 (1.3)	0
M111	4	111.1 & (4)	4 (1.3)	0
M218	4	218.1 (4)	3 (1.0)	1 (1.7)
AC4	3	229.0 (3)	2 (0.6)	0
AC5	3	31.10 *(3)	1 (0.3)	2 (3.5)
D1	2	207.0 (2)	2 (0.6)	0
M179	2	179.0 (2)	1 (0.3)	0
M26	1	26.3 (1)	1 (0.3)	0
M105	1	105.0 & (1)	1 (0.3)	0
Non-classified for *emm*-cluster	40	127.0 (1), 145.4 (1), 147.0 (2), 162.1 (4), 220.0 (2), 240.1 (1), STC46 (2), STG643.1 (2), STG866.1 (1), STG866.X2 *(1), STG7882.3 (1). Not able to be *emm*-typed (22)	28 (9.0)	6 (10.3)
		Not able to be *emm*-typed (22)		
		Not able to be *emm*-typed (22)		

* New *emm*-subtypes identified in this study. NINV = non-invasive, INV = invasive infections. # *emm*-type included in the 30-valent vaccine & emm-type cross-opsonized by 30-valent vaccine.

## Data Availability

Data are available at the MRCG at LSHTM and be made available upon request to the corresponding author.

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
