# Peer review of "Molecular Epidemiology of Group A Streptococcus Infections in The Gambia"

_vaccines, 2021, doi:10.3390/vaccines9020124_

Round 1

Reviewer 1 Report

The authors present the results of typing of Streptococcus isolates from Gambia, which delivers important information on the epidemiology of this infection as well as for the predictive evaluation of the 30-valent vaccine. Presented data adds to the current knowledge on genotypic diversity and invasiveness on the isolates in Gambia, puts these results in relation to similar findings for middle and less developed countries as well as developed countries, and with the discovery of novel S. pyogenes types („the untypeable types“) opens important questions regarding diagnosis and vaccine design for the future. To increase the interest of the broader audience and facilitate readers‘ comprehension, I would suggest to ammend the abstract with the information including basic characteristics of emm-typing (genotyping, improved set of primers for typing, etc). It would be of utter importance to expand on typing methodology briefly touched in the lines 76-79.

The Discussion section would be enriched with authors‘ views on incorporation of their findings into methodology of an alternative or improved vaccine design.

Minor remarks:

Line numbering starts again with 1 on Page 5?

Line 56: five countries

Line 88: 1 month

Line 109: please check the second part of the sentence for clarity

Author Response

You can find our point by point reply in the pdf file.

Reviewer 2 Report

This manuscript by Jabang and Erhart et al is a descriptive characterization of Group A Streptococcus (GAS) isolates from The Gambia between 2004 and 2018. This manuscript may be viewed as a Resource-type contribution with no further mechanistic studies performed. This is a useful contribution for understanding geographic stratifications of GAS in the global effort to develop a more effective GAS vaccine for low- and middle-income regions. The paper is succinct and well written, but I think that there is a missed opportunity to further describe the data rather than just reporting a table of emm-types. Despite the retrospective design, a finer analysis of some of the features of the data will provide even greater value to readers of this manuscript.

  1. Can the authors clarify more about whether patients from the MRCG Fajara site had comorbidities? i.e., if a patient had a skin infection, does that mean the patient did not have an ENT infection? The overlap of these clinical indicators is not clear. If there is no overlap in their stratification, it would be helpful to list that.
  2. As a follow-up to this, can the authors analyze their emm-clusters in the context of invasive for non-invasive infection? For example, E6 has 88 GAS isolates, for which 77 are stratified into 67 NINV and 10 INV. Which emm-types are associated with NINV and which are associated with INV?
  3. Further, can the authors contextualize the emm-types to clinical presentation? It will be interesting to know whether certain emm-types are associated with ENT infections, etc, for this specific geographic population.
  4. The authors indicate that 150 isolates were of emm-types covered by the 30-valent vaccine and 141 isolates were likely covered by cross-opsonization. Where is this result indicated? This is a valuable finding which is sort of glossed over by a single sentence. The authors should present this finding more fully, as it is highly relevant.

Author Response

(The authors gave the same response as above.)

Reviewer 3 Report

The manuscript by Erhart and co-workers describe the molecular epidemiology of selected infections in The Gambia. The manuscript is well presented with good introductions and discussion, even including limitations. 

Minor corrections:

  1. delete the extra space between The Florey in line 16.
  2. Use another for Although in line 37, since you have it on line 36.
  3. having N-terminal end is repetitive? see line 43
  4. line 64, different font for the link.
  5. another extra space in line 77 between an and emm
  6. A picture of an affected area will be of great addition to the manuscript. 
  7. since there are many authors, a section with each author contribution should be included 

Author Response

(The authors gave the same response as above.)

Round 2

Reviewer 2 Report

This is a revised manuscript by Jabang and Erhart et al on characterization of Group A Stroptococcus isolates from The Gambia. The authors have not satisfactorily addressed two of my four comments.

Comment 1: Regarding clinical presentation - it is understandable that there was no systematic collection of information on potential comorbidities. I have no further comments on this.

Comment 2: Regarding analyzing emm-clusters in the context of invasive or non-invasive infection. The authors have not addressed this question and simply refer me to the original line in the manuscript where they describe that there is a statistical significance in cluster distribution between invasive and non-invasive clusters. However, my question was asking the authors to specify which of the clusters were invasive or non-invasive. This information is obscured in their current analysis.

Comment 3: Regarding contextualizing emm-types to clinical presentation. It is understandable that there is insufficient clinical data to correlate these findings. I have no further comments on this.

Comment 4: Regarding which emm-types are covered by the 30-valent vaccine and which may be covered by cross-opsonization. The authors have not addressed this question and simply refer me to the original statement that I asked about. My question asked the authors to specify which of the emm-types are included in the 30-valent vaccine and which of the 141 other emm-types are expected be covered by cross-opsonization.

Author Response

Response to Reviewer 2 Comments

Comment 2: Regarding analyzing emm-clusters in the context of invasive or non-invasive infection. The authors have not addressed this question and simply refer me to the original line in the manuscript where they describe that there is a statistical significance in cluster distribution between invasive and non-invasive clusters. However, my question was asking the authors to specify which of the clusters were invasive or non-invasive. This information is obscured in their current analysis.

Answer to comment 2: There were no emm clusters specific to invasive (INV) or to non-invasive (NINV) infections, but their frequency were found to be significantly different. Indeed, among invasive infections, the most frequent cluster was E3 (24.1%) followed by E6 (17.2%), M95 (12.1%) and E2 and M55 (each 6.9%). On the other hand, among NINVs, the most frequent cluster was E6 (21.2%), while M95 (3.2%) and M55 (1.3%) were much less frequent. The full details of the frequency of each cluster among INV and NINV are shown in the table. 

We revised the related sentence line 110 - 112 which now reads as follows:

“Indeed, among invasive infections, the most predominant cluster was E3 (24.1%) followed by E6 (17.2%) then M95 (12.1%), E4 (8.9%), and E2 and M55 (each 6.9%) (Exact test for difference between INV-NINV, p-value=0.03)”.

 Comment 4: Regarding which emm-types are covered by the 30-valent vaccine and which may be covered by cross-opsonization. The authors have not addressed this question and simply refer me to the original statement that I asked about. My question asked the authors to specify which of the emm-types are included in the 30-valent vaccine and which of the 141 other emm-types are expected be covered by cross-opsonization.

Answer to comment 4: We have specified this information in table1. The 150 strains belonging to emm-types included in the 30-valent vaccine and the 141 strains belonging to cross-opsonised emm-types are indicated by “#” and “&” respectively. Thanks to the reviewer comment, we were able to identify typos in table 1. Such typos were corrected in the current version (see track changes), and we would like to express our gratitude for this.

Round 3

Reviewer 2 Report

The authors have satisfactorily responded to my comments.